# NOISE-AWARE ALGORITHM FOR HETEROGENEOUS DIFFERENTIALLY PRIVATE FEDERATED LEARNING

## ABSTRACT

Federated Learning (`FL`) is a useful paradigm for learning models from the data distributed among some clients. High utility and rigorous data privacy guaranties are among the main goals of an `FL` system. Previous works have tried to achieve the latter by ensuring differential privacy (`DP`) while performing federated training. In real systems, there is often heterogeneity in the privacy requirements of various clients, and the existing `DPFL` works either assume uniform requirements or propose methods relying on a trusted server. Furthermore, in real `FL` systems, there is also heterogeneity in memory/computing power across clients' devices, which has not been addressed in existing `DPFL` algorithms. Having these two sources of heterogeneity, straightforward solutions, e.g., meeting the privacy requirements of the most privacy-sensitive client or removing the clients with low memory budgets will lead to lower utility and fairness problems, due to high `DP` noise and/or data loss. In this work, we propose Robust-HDP to achieve high utility in the presence of an untrusted server, while addressing both the privacy and memory heterogeneity across clients. Our main idea is to efficiently estimate the noise in each client model update and assign their aggregation weights accordingly. Noise-aware aggregation of Robust-HDP without sharing clients privacy preferences with the server, results in the improvement of utility, privacy and convergence speed, while meeting the heterogeneous privacy/memory requirements of all clients. Extensive experimental results on multiple benchmark datasets and our convergence analysis confirm the effectiveness of Robust-HDP in improving system utility and convergence speed.

## 1 INTRODUCTION

Federated Learning (`FL`) [1] enables a set of clients to collaboratively train a machine learning (`ML`) model on their distributed data without data sharing. `FL` algorithms must be able to deal efficiently with clients' heterogeneous data and computational budgets to achieve high utility. Also, in the presence of sensitive information in the train data, `FL` algorithms must be able to provide rigorous data privacy guarantees against a potentially curious server or any third party [2–7]. Differential Privacy (`DP`) [8–11] provides such formal privacy guarantees. When there is a trusted server in the system, central differential privacy (`CDP`) [12, 13], which is operated by the trusted server by adding controlled noise to the aggregation of clients updates, has been proposed as a solution for achieving official data privacy in `FL`. Also, when there is no trusted server, which is more common, Local Differential Privacy (`LDP`), where each client runs the DPSGD [14] algorithm locally is a solution. However, `LDP` is limited in the sense that achieving privacy while preserving model utility is challenging, due to clients independent noise additions. Some solutions have been proposed for this, e.g., using a trusted shuffler system between clients and the untrusted server [15, 16], which may be difficult to establish if the server itself is untrsuted.

On the clients side, there are often heterogeneous privacy preferences, which can stem from their varying privacy policies. Also, depending on their computational budgets, some clients may not be capable of meeting high memory requirements of DPSGD algorithm [14], which needs to compute gradient vectors for each data sample in a batch of data separately. Such clients choose to use smaller batch sizes locally to meet their memory budgets, which we show analytically results in a fast increment of the noise level in their model updates. Existing heterogeneous `DPFL` works mostly depend on a trusted server [17, 18] or share clients sensitive information, e.g., privacy preferences, with an untrusted server [19]. While there are some works that are not related to `DPFL` [20–23]

and address the heterogeneous computational budgets across clients, there is no `DPFL` algorithm in the literature considering memory size (batch size) heterogeneity, which might be inevitable in `DPFL` systems. We consider heterogeneous `DPFL` systems with an untrusted server and propose an efficient algorithm with more focus on the client updates that are indeed less noisy. We propose to employ Robust PCA (RPCA) algorithm [24] by the untrusted server to estimate the amount of noise existing in the clients' model updates, which we show depends strongly on *multiple* factors (e.g., their privacy preference, their memory size budget (batch size) and dataset size), and assign their aggregation weights accordingly. The use of this efficient strategy on the server improves both the model utility and convergence speed. We confirm our understandings with both experimental results and theoretical analysis. The highlights of our contributions are summarized in the following:

- We show the considerable effect of batch size on the noise level in clients' model updates.
- We propose the efficient noise-aware "Robust-HDP" algorithm for heterogeneous `DPFL` with untrusted servers, which focuses on clients with less noisy model updates.
- As the first work assuming heterogeneous dataset sizes, heterogeneous batch sizes, non-uniform and varying aggregation weights and partial participation of clients simultaneously, we prove convergence of our proposed algorithm under mild assumptions on loss functions.
- We show the superiority of Robust-HDP in terms of higher utility, fast convergence and respecting clients' privacy compared to the existing heterogeneous `DPFL` algorithms.

## 2 RELATED WORK

In this work, we use the following as our main definition of `DP`:

**Definition 1** (($\epsilon, \delta$)-`DP` [9]). *A randomized mechanism $\mathcal{M} : \mathcal{D} \rightarrow \mathcal{R}$ with domain $\mathcal{D}$ and range $\mathcal{R}$ satisfies ($\epsilon, \delta$)-`DP` if for any two adjacent inputs $d, d' \in \mathcal{D}$, which differ in only one element, and for any measureable subset of outputs $\mathcal{S} \subseteq \mathcal{R}$ it holds that*

$$Pr[\mathcal{M}(d) \in \mathcal{S}] \leq e^{\epsilon} Pr[\mathcal{M}(d') \in \mathcal{S}] + \delta.$$

This definition captures the privacy guarantees of the Gaussian mechanism, which randomizes the output of a query $f$ on a dataset $d$, as: $\mathbf{G}_{\sigma} f(d) \triangleq f(d) + \mathcal{N}(0, \sigma^2)$. The randomized output of the Gaussian mechanism satisfies ($\epsilon, \delta$)-`DP` for a continuum of pairs ($\epsilon, \delta$): it is ($\epsilon, \delta$)-`DP` for all combinations of $\epsilon < 1$ and $\sigma > \frac{\sqrt{2 \ln(1.25/\delta)}}{\epsilon} \Delta_2 f$, where $\Delta_2 f \triangleq \max_{d \sim d'} \parallel f(d) - f(d') \parallel_2$ is the $l_2$-sensitivity of the query $f$, computed on neighboring datasets $d$ and $d'$. As an extension of ($\epsilon, \delta$)-`DP`, personalized `DP` (PDP), which specifies the privacy parameters for each sample in a dataset separately, was proposed for centralized settings [25–29]. Another similar work in [30] proposed "Utility Aware Exponential Mechanism" (UPEM) to pursue higher utility while achieving PDP. In the same direction of improving utility, Shi et al. [31] proposed "Selective `DP`" for improving utility by leveraging the fact that private information in natural language is sparse. Hence, providing rigorous privacy guarantees on the sensitive portion of data is enough and improves model utility.

**Heterogeneous Privacy Preference and Memory Size in `FL`:** The notion of heterogeneous `DP` has been extended to the `FL` setting as well, in which each client has its own desired privacy parameters ($\epsilon_i, \delta_i$). A naive approach to this problem is to design a uniform `DPFL` system to satisfy ($\epsilon_{\min}, \delta_{\min}$)-`DP`, where ($\epsilon_{\min}, \delta_{\min}$) comes from the most privacy sensitive client. However, this naive approach leads to a large amount of utility loss [12]. Assuming the existence of a *trusted* server, Chathoth et al. [17] proposed cohort-level privacy with privacy and data heterogeneity across cohorts using $\epsilon$-`DP` definition. Recently, the work in [18], adapted the non-uniform sampling idea of [26] to the `FL` settings with a trusted server to get client-level `DP` in `FL` for protection against membership inference attacks [3, 5]. In contrast, we are interested in the heterogeneous `DPFL` settings *with an untrusted server*, as it is more applicable.

Previous works [32–34] have found that stochastic gradients stay in a low-dimensional space during training process with Stochastic Gradient Descent (SGD) algorithm . Inspired by this observation, Zhou et al. [35] proposed projection-based variant of the DPSGD [14] algorithm (projected DPSGD) to achieve higher utility. Liu et al. [19] adapted the projection-based approach to the heterogeneous `DPFL` setting in order to reduce the noise in the model updates of privacy sensitive clients and

improve utility. It also used a naive aggregation strategy to assign aggregation weights proportional to privacy parameter $\epsilon$ of clients. Although [19] works with an untrusted server, *it is limited to cross-silo settings and relies on the assumption that the server knows the clients' privacy preferences* $\{\epsilon_i\}$ and uses them to cluster clients to "public" (those with largest privacy budgets) and "private." As $(\epsilon_i, \delta_i)$ generally represent the data sensitivity of client $i$, it might not always be possible to share them with an *untrusted* server. Furthermore, as we will show, even if the server knows clients' privacy parameters, this information is not necessarily indicative of the noise level in their model updates, especially when clients have different batch/dataset sizes,

To address the potential heterogeneity between clients' computational budgets [20, 36], some works have considered using different model sizes for clients, depending on their computational budgets, and used knowledge distillation techniques [37] to exchange knowledge between their heterogeneous models [20–22]. However these works are not proposed for DPFL settings. In contrast, we consider the same common model structure for all clients, but depending on their memory budgets, they use different batch sizes for their local training with the memory-consuming DPSGD algorithm [14].

The current state of the art calls for a heterogeneous DPFL algorithm that takes both privacy preference heterogeneity and batch size heterogeneity into account for achieving utility and data privacy.

**Notations** We consider an FL setting with $n$ clients. Let $x \in \mathcal{X} \subseteq \mathbb{R}^d$ and $y \in \mathcal{Y} = \{1, \ldots, C\}$ denote an input data point and its target label. Client $i$ holds dataset $\mathcal{D}_i$ with $N_i$ samples from distribution $P_i(x, y)$. Let $h : \mathcal{X} \times \boldsymbol{\theta} \to \mathbb{R}^C$ be the predictor function, which is parameterized by $\boldsymbol{\theta} \in \mathbb{R}^p$ shared among all clients. Also, let $\ell : \mathbb{R}^C \times \mathcal{Y} \to \mathbb{R}_+$ be the loss function used (cross entropy loss). Following [1], many existing FL algorithms fall into the natural formulation that minimizes the (arithmetic) average loss $f(\boldsymbol{\theta}) := \sum_{i=1}^n \lambda_i f_i(\boldsymbol{\theta})$, where $f_i(\boldsymbol{\theta}) = \frac{1}{N_i} \sum_{(x,y) \in \mathcal{D}_i} [\ell(h(x, \boldsymbol{\theta}), y)]$, with minimum value $f_i^*$. The weights $\boldsymbol{\lambda} = (\lambda_1, \ldots, \lambda_n)$ are nonnegative and sum to 1. At gradient update $t$, client $i$ uses a data batch $\mathcal{B}_i^t$ with size $b_i = |\mathcal{B}_i^t|$. Let $q_i = \frac{b_i}{N_i}$ be batch size ratio of client $i$. There are $E$ global communication rounds indexed by $e$, and in each of them, client $i$ runs $K_i$ local epochs.

## 3 THE Robust-HDP ALGORITHM FOR HETEROGENEOUS DPFL

In this section, we focus on devising a new heterogeneous DPFL algorithm and explain the motivations behind it. At the $t$-th gradient update step on a current model $\boldsymbol{\theta}$, client $i$ computes the following noisy batch gradient:

$$\tilde{g}_i(\boldsymbol{\theta}) = \frac{1}{b_i} \left[ \left( \sum_{j \in \mathcal{B}_i^t} \bar{g}_{ij}(\boldsymbol{\theta}) \right) + \mathcal{N}(0, \sigma_{i,\mathrm{dp}}^2 I_p) \right], \tag{1}$$

where $\bar{g}_{ij}(\boldsymbol{\theta}) = \mathtt{clip}(\nabla\ell(h(x_{ij}, \boldsymbol{\theta}), y_{ij}), c)$, $c$ is a clipping threshold while $\mathtt{clip}(v, c) = \min\{\|v\|, c\} \times \frac{v}{\|v\|}$), $\sigma_{i,\mathrm{dp}} = c \times z(\epsilon_i, \delta_i, q_i, K_i, E)$, and $z$ is the noise scale needed for achieving $(\epsilon_i, \delta_i)$−DP by client $i$, which can be determined with a privacy accountant, e.g., the moments accountant [14] used in this work. The question that we answer in this section is that what is the aggregation strategy that minimizes the noise in the aggregated model update on the server. To this end, we first analyze the effect of batch size on clients' batch gradients.

### 3.1 EFFECT OF BATCH SIZE ON CLIENTS' DP NOISY BATCH GRADIENTS

In this section, all the computations are conditioned on $\boldsymbol{\theta}$. Depending on the value of the used clipping threshold $c$ at the $t$-th gradient update step, we consider two general indicative cases:

**1. Effective clipping threshold:** when the clipping is indeed effective for all samples, we have:

$$\mathbb{E}[\tilde{g}_i(\boldsymbol{\theta})] = \frac{1}{b_i} \sum_{j \in \mathcal{B}_i^t} \mathbb{E}[\bar{g}_{ij}(\boldsymbol{\theta})] = \frac{1}{b_i} \sum_{j \in \mathcal{B}_i^t} G_i(\boldsymbol{\theta}) = G_i(\boldsymbol{\theta}), \tag{2}$$

where the expectation is with respect to the stochasticity of gradients and we have assumed that $E[\bar{g}_{ij}(\boldsymbol{\theta})]$ is the same for all $j$ and is denoted by $G_i(\boldsymbol{\theta})$. Also, the variance of the noisy stochastic

gradient in Equation (1) can be computed as (see Appendix B.1):

$$\sigma^2_{i,\tilde{g}} := \mathtt{Var}(\tilde{g}_i(\boldsymbol{\theta})) = \frac{c^2 - \left\|G_i(\boldsymbol{\theta})\right\|^2}{b_i} + \frac{pc^2 z^2(\epsilon_i, \delta_i, q_i, K_i, E)}{b_i^2} \approx \frac{pc^2 z^2(\epsilon_i, \delta_i, q_i, K_i, E)}{b_i^2}. \quad (3)$$

**2. Ineffective clipping threshold:** when the clipping is ineffective for all samples, we have a noisy version of the batch gradient $g_i(\boldsymbol{\theta}) = \frac{1}{b_i} \sum_{j \in \mathcal{B}_i^t} g_{ij}(\boldsymbol{\theta})$, which is unbiased with variance bounded by $\sigma^2_{i,g}$ (see Assumption 1). Hence:

$$\mathbb{E}[\tilde{g}_i(\boldsymbol{\theta}] = \mathbb{E}[g_i(\boldsymbol{\theta})] = \nabla f_i(\boldsymbol{\theta}), \quad (4)$$

$$\sigma^2_{i,\tilde{g}} = \mathtt{Var}(\tilde{g}_i(\boldsymbol{\theta})) = \mathtt{Var}(g_i(\boldsymbol{\theta})) + \frac{p\sigma^2_{i,\mathrm{dp}}}{b_i^2} \le \sigma^2_{i,g} + \frac{pc^2 z^2(\epsilon_i, \delta_i, q_i, K_i, E)}{b_i^2}. \quad (5)$$

Hence, we observe that, as $z$ grows with $b_i$ *sub-linearly* (see Figure 6 in the appendix), $\mathtt{Var}(\tilde{g}_i(\boldsymbol{\theta}))$ *is a decreasing function of $b_i$*. This means that *the lower the batch size of a client (which represents its device memory size), the larger the noise it adds to its batch gradients for achieving the same level of privacy. Also, the lower the privacy budget $(\epsilon, \delta)$ of a client, the larger the noise it adds.*

## 3.2 EFFECT OF BATCH SIZE ON CLIENTS' NOISY MODEL UPDATES

We now investigate the effect of batch size on the noise level in clients' model updates. During each global communication round, a participating client $i$ performs $E_i = K_i \times \lceil \frac{N_i}{b_i} \rceil = K_i \times \lceil \frac{1}{q_i} \rceil$ batch gradient updates locally with step size $\eta_l$:

$$\Delta\tilde{\boldsymbol{\theta}}_i^e = \boldsymbol{\theta}_{i,E_i}^e - \boldsymbol{\theta}_{i,0}^e,$$
$$\boldsymbol{\theta}_{i,k}^e = \boldsymbol{\theta}_{i,k-1}^e - \eta_l \tilde{g}_i(\boldsymbol{\theta}_{i,k-1}^e), \ k = 1, \ldots, E_i. \quad (6)$$

In each update, it adds a Gaussian noise from $\mathcal{N}(0, \frac{c^2 z^2(\epsilon_i, \delta_i, q_i, K_i, E)}{b_i^2} I_p)$ to its batch gradients independently (see Equation (1)). Hence:

$$\sigma_i^2 := \mathtt{Var}(\Delta\tilde{\boldsymbol{\theta}}_i^e | \boldsymbol{\theta}^e) = K_i \times \lceil \frac{1}{q_i} \rceil \times \eta_l^2 \times \sigma^2_{i,\tilde{g}}, \quad (7)$$

where $\sigma^2_{i,\tilde{g}}$, for two general indicative cases, was computed in Equation (3) and Equation (5). This means that $\sigma_i^2$ heavily depends on $b_i$ (e.g. when clipping is effective, $b_i$ appears with power 3 in denominator (recall $\frac{1}{q_i} = \frac{N_i}{b_i}$).

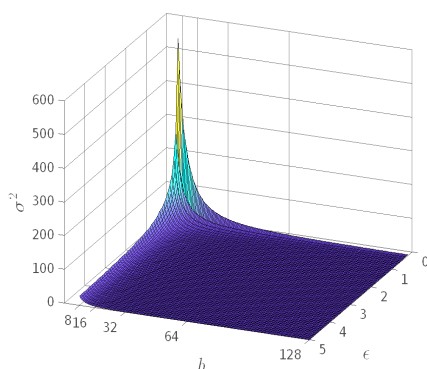

Figure 1: 3D plot of noise variance (eq. (7) and eq. (3) with $K = 1, N = 2400, \eta_l = 0.01, c = 3, p = 28939$) based on batch size ratio $q$ and the privacy budget $\epsilon$.

Hence, $\sigma_i^2$ *decreases quickly when $b_i$ increases. Equivalently, as the memory size of a client's device decreases, its model updates get noisier quickly. Note that, this scenario repeats when all clients use the same batch size but they have different dataset sizes, which results in different batch size ratios ($\{q_i\}_{i=1}^n$). Generally, a client with small privacy parameter and small batch size ratio ($q_i$) sends more noisy updates, compared to other clients (see Figure 1).*

Assuming that the set of clients $\mathcal{S}^e$ are participating in round $e$, and at the end of this round, the server assigns an aggregation weight $w_i$ to client $i$'s update $\Delta\tilde{\boldsymbol{\theta}}_i^e$, in order to minimize the total noise in the aggregated model update, we solve the following problem:

$$\min_{w_i \ge 0} \ \mathtt{Var}\left( \sum_{i \in \mathcal{S}^e} w_i \Delta\tilde{\boldsymbol{\theta}}_i^e | \boldsymbol{\theta}^e \right) = \sum_{i \in \mathcal{S}^e} w_i^2 \sigma_i^2, \qquad \mathtt{s.t.} \sum_{i \in \mathcal{S}^e} w_i = 1, \quad (8)$$

which has a unique solution $w_i^* \propto \frac{1}{\sigma_i^2}$, i.e. $w_i^* = \frac{1/\sigma_i^2}{\sum_{j \in \mathcal{S}^e} 1/\sigma_j^2}$. Hence, the optimum aggregation strategy to minimize noise on the server, weights clients directly based on $\{\sigma_i^2\}_{i=1}^n$, which in turn, *not only depends on $\{\epsilon_i\}_{i=1}^n$, but it also depends on $\{b_i\}_{i=1}^n$ and $\{N_i\}_{i=1}^n$ too. Therefore, assigning the aggregation weights $w_i^e = \frac{\epsilon_i}{\sum_j \epsilon_j}$ (as in [19] for* PFA *and* WeiAvg *algorithms), may be suboptimal*

Table 1: Features of different heterogeneous `DPFL` algorithms. ×: needed at server, ✓: not needed.

| algorithm | aggregation strategy | $\{\epsilon_i\}_{i=1}^n$ | clustering | projection of clients updates |
|---|---|---|---|---|
| WeiAvg [19] | $w_i \propto \epsilon_i$ | × | × | ✓ |
| PFA [19] | $w_i \propto \epsilon_i$ | × | × | × |
| DPFedAvg [38] | $w_i \propto N_i$ | ✓ | ✓ | ✓ |
| minimum $\epsilon$ | $w_i \propto N_i$ | × | ✓ | ✓ |
| Robust-HDP | $w_i \propto \frac{1}{\sigma_i^2}$ | ✓ | ✓ | ✓ |

*and inefficient*. On the other hand, when the server is not to be trusted, it usually does not have any idea of the clients noise addition mechanisms, their $\sigma_i^2$ or even their corresponding privacy parameters $(\epsilon_i, \delta_i)$, as they are kept private by clients. For instance, a small $(\epsilon_i, \delta_i)$ indicates greater willingness to sccrifice utility and may reveal the sensitivity of this client's data to a curious server, prompting unwanted attention. This is an important point that limits the applicability of the algorithms in [19], which heavily rely on the server knowing $\{(\epsilon_i, \delta_i)\}_{i=1}^n$, let alone their extra computational cost of clustering and denoising clients' updates using PCA (see Table 1). In the next section, we propose our idea for estimating $\{\sigma_i^2\}_{i=1}^n$ and $\{w_i^*\}_{i=1}^n$, *without knowing* $\{(\epsilon_i, \delta_i)\}_{i=1}^n$ on the server side.

### 3.3 Robust-HDP: AN EFFICIENT AGGREGATION STRATEGY FOR HETEROGENEOUS DPFL

We now explain our proposed algorithm for heterogeneous `DPFL`, Robust-HDP, which has roots in the Robust Principal Component Analysis (RPCA) [24]. Consider a `FL` setting with $n$ clients. Assuming the same $\delta$ for all clients for simplicity, client $i$ desires $(\epsilon_i, \delta)$-`DP`. Based on its local memory size, each client $i$ determines its local batch size $b_i$. Having determined $E$ number of global communication rounds and its desired privacy parameter $(\epsilon_i)$, each client computes its required noise scale $z_i$ using e.g. moments accountant [14]. At the end of each global round $e$, the server gets the matrix $\mathbf{M}$, as shown below. As all the clients are training the same shared model on more or less similar data distributions, we expect $\mathbf{M}$ to have a low rank, based on the findings in [32–34]. On the other hand, as $\mathbf{M}$ is a low rank matrix with noisy columns, we can use "Robust Principal Component Analysis (RPCA)" [24] and decompose it as follows:

$$\mathbf{M} := [\Delta\tilde{\boldsymbol{\theta}}_1^e| \dots |\Delta\tilde{\boldsymbol{\theta}}_n^e] = \mathbf{L} + \mathbf{S},$$

where $\mathbf{L}$ is a low rank matrix and $\mathbf{S}$ is an sparse noise matrix. Based on how RPCA works ([24], see Algorithm 3 in the appendix), $\mathbf{L}$ is a low rank matrix (see Figure 8 in the appendix) estimating the "true" values of clients' updates and "$\mathbf{S}$" captures the noises induced by two sources: `DP` additive Gaussian noise and batch gradients stochastic noise, which lie outside the low rank subspace of model updates. Hence, we can use $\hat{\sigma}_i^2 := \frac{\|\mathbf{S}_{:,i}\|_2^2}{p}$ ($\mathbf{S}_{:,i}$ is the $i$-th column of $\mathbf{S}$) as an estimate of the average amount of noise in $\Delta\tilde{\boldsymbol{\theta}}_i^e$. Thus, according to Eq. 8, we assign the aggregation weights as $w_i^e = \frac{1/\hat{\sigma}_i^2}{\sum_{j \in \mathcal{S}^e} 1/\hat{\sigma}_j^2}$, where $\hat{\sigma}_i^2 = \frac{\|\mathbf{S}_{:,i}\|^2}{p}$. Algorithm 1 summarizes our proposed Robust-HDP.

### 3.4 RELIABILITY OF Robust-HDP

How reliable is Robust-HDP for returning the optimal weights $\{w_i^*\}$? Note that, based on that $w_i^* \propto \frac{1}{\sigma_i^2}$, *in order for* Robust-HDP *to assign the optimum aggregation weights* $\{w_i^*\}$, *it suffices to estimate the set* $\{\sigma_i^2\}$ *up to a multiplicative factor*. Hence, we have the following lemma.

**Lemma 1** (**Precision of** Robust-HDP). *Let $s_{i,j}$ in matrix $\mathbf{S}$ represent the true value of noise in the $i$-th element of $\Delta\tilde{\boldsymbol{\theta}}_j^e$ ($j \in \mathcal{S}^e$). Then, assume that $\mathbf{S}'$ is the matrix computed by* Robust-HDP *at the server with bounded elements $s_{i,j}'^2 \leq U$, where $\mathbb{E}[s_{i,j}'] = rs_{i,j}$, for some constant $r > 0$, and $\mathbb{E}[|s_{i,j}' - rs_{i,j}|^2] \leq \alpha_j^2$ (i.e., on average,* Robust-HDP *is able to estimate the true noise values $s_{i,j}$ up to a multiplicative factor $r$). Then:*

$$Pr(|\hat{\sigma}_j^2 - (r^2\sigma_j^2 + \alpha_j^2)| > \epsilon) \leq 2e^{\frac{-2p\epsilon^2}{U^2}}. \tag{9}$$

This means that estimating the elements in the true noise matrix $\mathbf{S}$ up to a multiplicative factor $r$ with a small variance is enough for Robust-HDP to get to a close estimate of the true noise variances $\{\sigma_i^2\}$

**Algorithm 1:** Robust-HDP

**Input:** Initial parameter $\boldsymbol{\theta}^0$, batch sizes $\{b_1, \ldots, b_n\}$, dataset sizes $\{N_1, \ldots, N_n\}$, noise scales $\{z_1, \ldots, z_n\}$, gradient norm bound $c$, local epochs $\{K_1, \ldots, K_n\}$, global round $E$, number of model parameters $p$, privacy accountant **PA**.

**Output:** $\boldsymbol{\theta}^E, \{\epsilon_1^E, \ldots, \epsilon_n^E\}$

1 **Initialize** $\boldsymbol{\theta}_0$ randomly
2 **for** $e \in [E]$ **do**
3      sample a set of clients $\mathcal{S}^e \subseteq \{1, \ldots, n\}$
4      **for** *each client $i \in \mathcal{S}^e$ **in parallel*** **do**
5          $\Delta\tilde{\boldsymbol{\theta}}_i^e \leftarrow$ **DPSGD**$(\boldsymbol{\theta}^e, b_i, N_i, K_i, z_i, c)$
6          $\epsilon_i^e \leftarrow$ **PA**$(\frac{b_i}{N_i}, z_i, K_i, e)$
7      $\mathbf{M} = [\Delta\tilde{\boldsymbol{\theta}}_1^e | \ldots | \Delta\tilde{\boldsymbol{\theta}}_{|\mathcal{S}^e|}^e] \in \mathbb{R}^{p \times |\mathcal{S}^e|}$
8      $\mathbf{L}, \mathbf{S} = $ **RPCA**$(\mathbf{M})$
9      **for** $i \in \mathcal{S}^e$ **do**
10          $w_i^e \leftarrow \frac{1/\|\mathbf{S}_{:,i}\|_2^2}{\sum_{j \in \mathcal{S}_e} 1/\|\mathbf{S}_{:,j}\|_2^2}$
11      $\boldsymbol{\theta}^{e+1} \leftarrow \boldsymbol{\theta}^e + \sum_{i \in \mathcal{S}_e} w_i^e \Delta\tilde{\boldsymbol{\theta}}_i^e$

up to a multiplicative factor $r^2$ with high probability. This probability increases with the number of model parameters $p$ exponentially. Also, note that $w_j \propto \frac{1}{\hat{\sigma}_j^2}$. Hence, as $\sigma_j^2 \gg 1$ (it is the noise power in the whole model parameter vector with length $p$), a small deviation $\alpha_j^2$ from $r^2 \sigma_j^2$ still results in the Robust-HDP returning aggregation weights close to the optimum weights $\{w_i^*\}$.

### 3.5 SCALABILITY OF Robust-HDP WITH THE NUMBER OF MODEL PARAMETERS $p$

The computation time (precision) of RPCA algorithm increases (decreases) when the number of model parameters $p$ grows. For instance, we use a much larger model for CIFAR10, compared to MNIST and FMNIST. As such, in order to make the Robust-HDP scalable for large models, we perform the noise estimation mechanism of Robust-HDP on sub-matrices of $\mathbf{M}$ (e.g., $\mathbf{M}[0:p',:] = \mathbf{L} + \mathbf{S}$) in parallel and use their average noise variance estimate for weight assignment. For instance, for CIFAR-10, we perform RPCA on sub-matrcies of $\mathbf{M}$ with $p' = 200,000$ rows, and average their noise variance estimates. Experimental results show that this approach still results in assigning aggregation weights close to the optimum weights $\{w_i^*\}$ (as observed in Figure 9 in the appendix for CIFAR-10). This idea makes Robust-HDP scalable to large models with high number of parameters.

### 3.6 PRIVACY ANALYSIS OF Robust-HDP

We have the following theorem about DP guarantees of our proposed Robust-HDP algorithm.

**Theorem 1.** *For each client $i$, there exist constants $c_1$ and $c_2$ such that given its number of steps $E \cdot E_i$, for any $\epsilon < c_1 q_i^2 E \cdot E_i$, the output model of* Robust-HDP *satisfies $(\epsilon_i, \delta_i)-$DP with respect to $\mathcal{D}_i$ for any $\delta_i > 0$ if $z_i > c_2 \frac{q_i \sqrt{E \cdot E_i \log \frac{1}{\delta_i}}}{\epsilon_i}$, where $z_i$ is the noise scale used by the client $i$ for DPSGD. The algorithm also satisfies $(\epsilon_{max}, \delta_{max})$-DP, where $(\epsilon_{max}, \delta_{max}) = \big(\max(\{\epsilon_i\}_{i=1}^n), \max(\{\delta_i\}_{i=1}^n)\big)$.*

Therefore, the model returned by Robust-HDP is $(\epsilon_i, \delta_i)$-DP with respect to $\mathcal{D}_i$, meaning that Robust-HDP satisfies clients heterogeneous privacy preferences.

### 3.7 THE OPTIMIZATION SIDE OF Robust-HDP

We assume that $f(\boldsymbol{\theta}) = \sum_{i \in [n]} \lambda_i f_i(\boldsymbol{\theta})$, where $\lambda_i = \frac{N_i}{\sum_i N_i}$, has minimum value $f^*$ and minimizer $\boldsymbol{\theta}^*$. We also make some mild assumptions about the loss functions $f_i$ (see Assumption 1 and the notations used therein). We are ready to analyze the convergence of the Robust-HDP algorithm.

**Theorem 2** (Robust-HDP). *Assume that Assumption 1 holds, and for every $i$, learning rate $\eta_l$ satisfies:* $\eta_l \leq \frac{1}{6\beta E_i}$ *and* $\eta_l \leq \frac{1}{12\beta\sqrt{\left(1+\sum_{i=1}^{n} E_i\right)\left(\sum_{i=1}^{n} E_i^4\right)}}$. *Then, we have:*

$$\min_{0 \leq e \leq E-1} \mathbb{E}[\|\nabla f(\boldsymbol{\theta}^e)\|^2] \leq \frac{12}{(11E_l^{min} - 7)}\left(\frac{f(\boldsymbol{\theta}^0) - f^*}{E\eta_l} + \frac{\sum_{e=0}^{E-1}(\Psi_\sigma^e + \Psi_p^e)}{E}\right), \quad (10)$$

*where $E_l^{min} = \min_i E_i$ and*

$$\Psi_\sigma^e = 6\beta^2\eta_l^2(1 + \sum_{i=1}^{n} E_i)\left(2\sum_{i=1}^{n} E_i^4\sigma^2 + \frac{1}{3}\sum_{i=1}^{n} E_i^3\sigma_{i,\tilde{g}}^2\right) + \beta\eta_l\sum_{i=1}^{n} E_i^2\sigma_{i,\tilde{g}}^2$$

$$\Psi_p^e = \frac{8L_0^2}{3}\left(n\sum_{i=1}^{n} E_i^2\mathbb{E}[(w_i^e - \lambda_i)^2] + \|\boldsymbol{\lambda}\|^2\sum_{i=1}^{n} \mathbb{E}[(E_i - \mu_w^e)^2]\right), where \ \mu_w^e = \sum_{i=1}^{n} w_i^e E_i. \quad (11)$$

**Discussion.** Our convergence guarantees are quite general: we allow for partial participation, heterogeneous number of local steps $\{E_i\}$, non-uniform batch sizes $\{b_i\}$, varying and nonuniform aggregation weights $\{w_i^e\}$. When $\{f_i\}$ are convex, Robsut-HDP solution converges to a neighborhood of the optimal solution. The term $\Psi_\sigma$ decreases when $E_i$ and variance of mini-batches $\{\sigma_{i,\tilde{g}}^2\}$ decrease (e.g., when clients are less privacy sensitive and have larger memory sizes). Similarly, $\Psi_p$ decreases when clients participate more often and have similar number of data samples and use similar batch sizes. Compared to the results in previous DPFL works, we have the most general results with more realistic assumptions. For instance, [19] (WeiAvg and PFA algorithms) assumes uniform number of local SGD updates for all clients, or [38] (DPFedAvg algorithm) assumes uniform aggregation weights and uniform number of local updates. These assumptions may not be practical in real systems. In a more general view, when we have no DP guarantees, we recover the results for the simple FedAvg algorithm [39]. When we additionally have $\sigma = 0$ (i.e., FedAvg on iid data), our results is the same as the results of stochastic gradient descent [40], since it reduces to:

$$\min_{0 \leq e \leq E-1} \mathbb{E}[\|\nabla f(\boldsymbol{\theta}^e)\|^2] \leq \frac{12}{(11E_l^{\min} - 7)}\frac{f(\boldsymbol{\theta}^0) - f^*}{E\eta_l} + \mathcal{O}(\eta_l), \quad (12)$$

which shows convergence rate $\frac{1}{\sqrt{E}}$ with $\eta_l = \mathcal{O}(\frac{1}{\sqrt{E}})$.

## 4 EXPERIMENTS

We now perform comprehensive experiments to evaluate our proposed Robust-HDP algorithm in terms of test accuracy, its precision in assigning optimum aggregation weights and also its convergence speed. See appendix A for details of experimental setup and hyperparameter tuning.

### 4.1 EXPERIMENTAL SETUP

**Datasets, models and baseline algorithms:** We evaluate our proposed method on three benchamrk datasets: MNIST [41], FMNIST [42] and CIFAR-10 [43] using CNN-based models (see Appendix A). Also, we compare four baseline algorithms: 1. WeiAvg [19]: weighted average aggregation based on privacy parameters $\{\epsilon_i\}_{i=1}^n$ 2. PFA [19]: WeiAvg with PCA projection of noisy model updates 3. DPFedAvg [38]: FedAvg with DPSGD 4. minimum $\epsilon$: FedAvg with uniform $(\epsilon_{\min}, \delta_{\min})$-DP

**Privacy preference and batch size heterogeneity:** We consider an FL setting with 20 clients and full participation and one local epoch for each client ($K_i = 1$ for all $i$). Due to data and batch size heterogeneity, the number of local steps $E_i$ for each client $i$ varies. We simulate the privacy preference heterogeneity across clients by sampling $\{\epsilon_i\}_{i=1}^n$ from different distributions, as shown in Table 4 in the appendix. We also sample the clients batch sizes $\{b_i\}_{i=1}^n$ uniformly from $\{16, 32, 64, 128\}$.

### 4.2 EXPERIMENTAL RESULTS

In this section, we investigate four main research questions based on our experimental results.

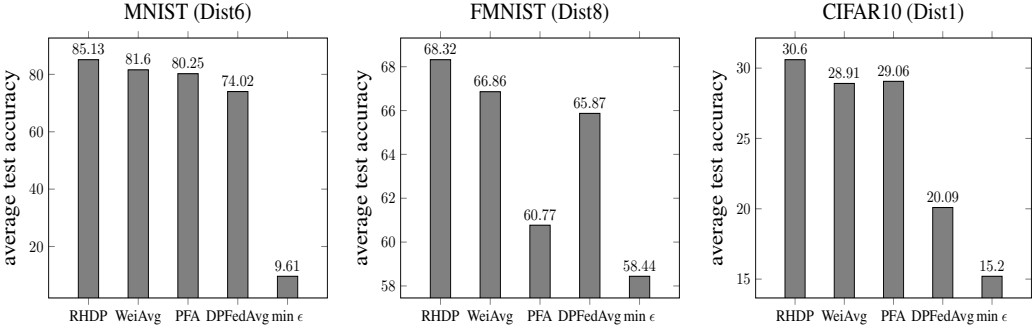

Figure 2: Utility comparison between Robust-HDP and the baseline algorithms. See Tables 8, 9 and **??** in the appendix for detailed results, including standard deviation over three independent runs.

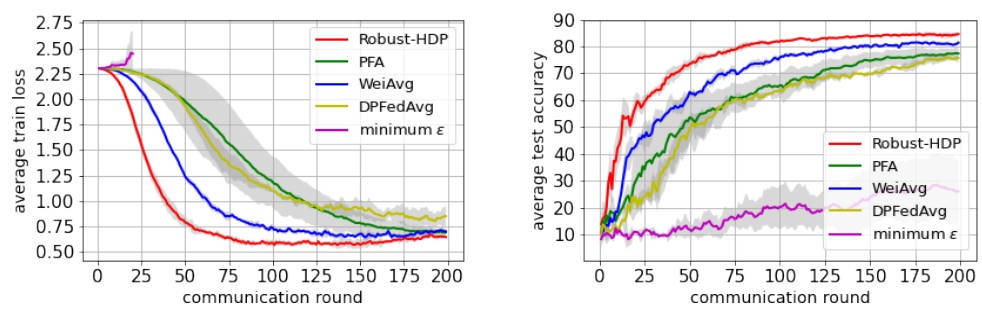

Figure 3: Convergence speed comparison on MNIST and Dist6.

**RQ1: How do various heterogeneous `DPFL` algorithms affect the utility of an `FL` system?** In Fig. 2, we have compared our proposed algorithm with others in terms of the average test accuracy across clients. We observe that Robust-HDP outperforms the baselines (detailed results for all privacy distributions are reported in tables 8, 9 and **??** in the appendix). Robust-HDP assigns smaller weights to the noisier updates, and when clients are more privacy sensitive (e.g. in Dist9), this results in a larger utility gap between Robust-HDP and others. Another point observed from the results is that, unlike what claimed in [19], projection of noisy clients updates on less noisy ones by PFA algorithm, did not result in utility improvement compared to WeiAvg in our experiments. We think this is because neural networks are often sensitive to their model parameters, while projection of model updates using PCA replaces them with another set of parameters.

**RQ2: How does Robust-HDP improve convergence speed during training?** We have also compared different algorithms based on their convergence speed in Figure 3. While the baseline algorithms suffer from high levels of noise in the aggregated model update $\sum_{i \in \mathcal{S}^e} w_i^e \Delta \tilde{\boldsymbol{\theta}}_i^e$ (see Table 2; also Table 10 in the appendix), Robust-HDP enjoys its efficient noise minimization, which performs very close to the optimum aggregation strategy, and not only results in faster convergence but also improves utility. In contrast, based on our experiments, the baseline algorithms have to use smaller learning rates to avoid divergence of their training optimization. Note that *Fast convergence of `DPFL` algorithms is indeed important, as the privacy budgets of participating clients does not let the server to run the federated training for more number of communication rounds.*

**RQ3: How accurate Robust-HDP is in estimating $\{\sigma_i^2\}$?** Figure 4 compares the noise variance estimated by Robust-HDP with their true values (computed from Equations 7 and 3) for MNIST dataset and Dist8. We have also compared the weight assignments in Figure 10 in the appendix. Figure 4 has sorted the clients based on their privacy parameter $\epsilon$. WeiAvg and PFA assign smaller weights to more privacy sensitive clients, while Robust-HDP assigns smaller weights to the clients based on the level of noise in their updates, which not only depends on their privacy parameter $\epsilon$, but also heavily on their batch size. If a client uses a large batch size (e.g. client 10 with batch size 128 in Figure 10 in the appendix), it is assigned a much larger weight, because of its much smaller noise variance. The baseline algorithms can not take this very important point into account at aggregation time, *even if they share privacy parameters $\{\epsilon_i\}$ with an untrusted server.*

Table 2: The average noise power (eq. (7) and eq. (3)) in each parameter normalized by used learning rate ($\frac{\sum_{i=1}^{n} w_i^{e\,2} \sigma_i^2}{p \eta_l^2}$) in the aggregated model updates for different algorithms (on FMNIST with $E = 200$). Due to the projection used in PFA, computing its noise power was not possible.

| dist / alg | Dist1 | Dist2 | Dist3 | Dist4 | Dist5 | Dist6 | Dist7 | Dist8 | Dist9 |
|---|---|---|---|---|---|---|---|---|---|
| WeiAvg [19] | 1.02 | 1.89 | 0.92 | 3.22 | 4.58 | 28.29 | 9.85 | 48.15 | 34.91 |
| DPFedAvg [38] | 1.27 | 16.94 | 16.28 | 26.87 | 25.64 | 70.71 | 18.50 | 85.70 | 43.20 |
| minimum $\epsilon$ [19] | 4.68 | 103.91 | 103.91 | 127.18 | 103.91 | 1868.45 | 74.41 | 241.37 | 87.15 |
| Robust-HDP | 0.27 | 0.47 | 0.07 | 0.64 | 0.39 | 7.62 | 2.25 | 13.86 | 5.94 |
| Oracle (eq. (8)) | 0.27 | 0.47 | 0.07 | 0.64 | 0.39 | 7.60 | 2.25 | 13.81 | 5.93 |

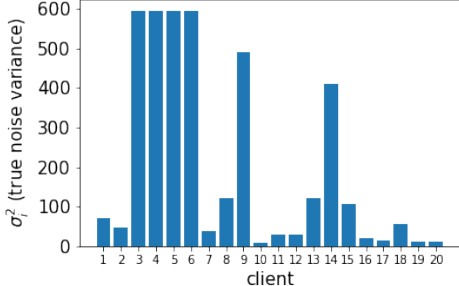 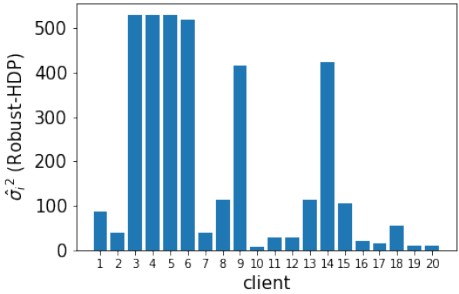

Figure 4: Comparison of $\{\sigma_i^2\}$ and their estimates $\{\hat{\sigma}_i^2\}$ from Robust-HDP, for MNIST and Dist8.

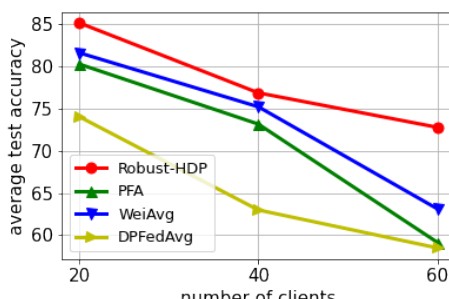 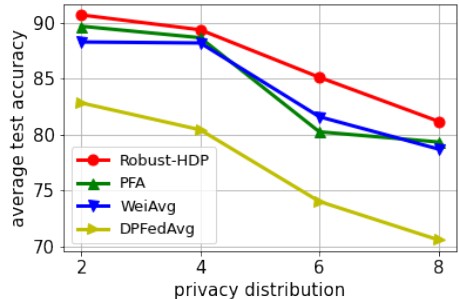

Figure 5: Performance comparison on MNIST dataset. **Left:** effect of clients desired privacy level. **Right:** effect of number of existing clients (privacy parameters of clients are sampled from Dist6). Due to its poor performance, we have not shown the results for "minimum $\epsilon$" for better visibility.

**RQ4: Is Robust-HDP indeed Robust?** In Fig. 5, we compare the test accuracy obtained from Robust-HDP with others based on the clients desired level of privacy and number of clients. As clients become more privacy sensitive, they send more noisy updates to the server, making convergence to better solutions harder. Robust-HDP shows the highest robustness to the larger noise in clients updates and achieves the highest utility, especially in more privacy sensitive scenarios, e.g., Dist8. Also, we observe that it achieves the highest system utility when the number of clients in the system increase, while performance of WeiAvg and PFA get close to that of DPFedAvg.

## 5 CONCLUSION

In heterogeneous DPFL systems, heterogeneity in privacy preference and memory size results in large variations in the noise levels across clients model updates. To address this heterogeneity, we proposed an efficient heterogeneous DPFL algorithm to perform noise-aware aggregation on an untrusted server without sharing clients' privacy parameters. Our theoretical and experimental results confirm that our algorithm results in better utility and faster convergence, while respecting clients' privacy.

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
