# OpenReview forum: "Noise-Aware Algorithm for Heterogeneous Differentially Private Federated Learning"
_ICLR.cc/2024/Conference — ICLR 2024 Conference Withdrawn Submission_

### Official Review · Reviewer_eRk3 · 2023-10-26

**Soundness:** 4 excellent
**Presentation:** 4 excellent
**Contribution:** 4 excellent
**Rating:** 8
**Confidence:** 3

**Summary:**

This paper proposes a novel aggregation method, Robust-HDP, for local differential privacy (LDP) in federated learning (FL) to attain better utility and convergence with heterogeneous clients. More specifically, Robust-HDP can handle different privacy requirements on different clients and assign the best aggregation weights for clients with different noisy levels of LDP. Theoretical and empirical results verify the effectiveness of Robust-HDP.

**Strengths:**

This work tries to address a practical and important issue in LDP that heterogeneous hardware and privacy requirements may significantly impact the utility and convergence of federated learning. The proposed Robust-HDP can handle the different noise levels in the uploaded models from different clients and assign the optimal weights to these models. The advantages are:

1.	Robust-HDP can evaluate the noise level by robust PCA without accessing the privacy requirements of each client, which leads to better privacy protection. The experimental results show that the evaluation method is accurate.

2.	Privacy and convergence are guaranteed by the theoretical analysis, while extensive experiments reveal the effectiveness of Robust-HDP.

3.	The paper is organized in a logical way, by justifying each design in the methodology with both theoretical analysis and empirical evaluation. This makes the paper comprehensible and solid.

**Weaknesses:**

Some possible points to further improve this paper are:

1.	A client with a larger dataset will have larger $E_i$ and thus higher noise. In this case, this client may be assigned a smaller weight $w_i$, which may violate the fairness requirements and lead to a biased global model. Though this paper is mainly focusing on privacy, fairness issues can be discussed.

2.	The experimental results shown in the body part are mainly on MNIST and FMNIST. Since CIFAR10 is more difficult and closer to a realistic setting, it would be preferable to show the ablation study results on CIFAR10 in the body. Also, the results of CIFAR10 seem to be lost in the Appendix.

**Questions:**

1.	I do not find how the data heterogeneity is simulated in the experiments. Could you provide more information about this part?

2.	Since the server can find the noise added to each local model update in $S$, the server can eliminate the noise from the model and get the noise-free results of the local model update. Does it violate the privacy requirements in LDP?

---

> ### Author Response · Authors · 2023-11-22
> **Response to Reviewer eRk3**
>
> Thank you for your feedbacks and comments about our work. We are happy to answer your comments and questions:
> > **Comment1**: A client with a larger dataset will have larger $E_i$ and thus higher noise.
>
> **A1**: The equation for $\sigma_i^2$ can be found in eq. 7, 5 and 3, where $N_i$ appears twice  with opposite effects. Hence $\sigma_i^2$ varies very slowly with $N_i$ (we have experimentally confirmed this). This means that the noise variance $\sigma_i^2$ of a client is mainly determined by its privacy parameter $\epsilon_i$ and batch size $b_i$. So if two clients have similar $\epsilon_i$ and $b_i$, the aggregation weight assigned to them by Robust-HDP is very close, even if their dataset sizes are different. In contrast, DPFedAvg assigns lower weights to clients with smaller dataset sizes. **As all clients have the same  $\sigma_i^2$ in this case, it makes sense to assign larger weights to the clients with larger dataset sizes (as DPFedAvg does) to improve utility**. But what happens to fairness in the system? it has been shown that accuracy of DP models drops much more for the underrepresented subgroups [1].  Hence, in some scenarios, we expect that Robust-HDP improves fairness in the system for clients with lower $N_i$. For instance, the following results evaluate the utility and also fairness in the system when we have homogeneous DPFL and only $N_i$ varies across clients:
>
> [1]: E. Bagdasaryan, "Differential Privacy Has Disparate Impact on Model Accuracy", arXiv 2019.
>
>
> * utility (average test accuracy). Note that PFA and WeiAvg are equivalent when $\\{\epsilon_i\\}_i$ is uniform:
> | Algorithm                                            |  $\epsilon=2.6$  |  $\epsilon=2$ |  $\epsilon=1.1$  | $\epsilon=0.6$  | $\epsilon=0.35$ |
> |-------------------------------------------|--------------------|-----------------|--------------------|--------------------|--------------------|
> | WeiAvg and PFA                                 |  37.24                 | **34.90**        |  27.80                 | 23.22                  | 19.01                  |
> | DPFedAvg  and minimum $\epsilon$ |  **38.52**            | 32.78             |  **30.42**            | **23.50**             | 20.26                 |
> | Robust-HDP                                        |  37.68.               | 32.54.            |  27.51                 | 22.58                   | **20.52**            |
>
>
> * performance fairness, measured by std of clients test accuracies and also the test accuracy of the client with smallest dataset size ($N_i$) (in parentheses):
> | Algorithm                                            |  $\epsilon=2.6$    |  $\epsilon=2$     |  $\epsilon=1.1$ | $\epsilon=0.6$        | $\epsilon=0.35$ |
> |-------------------------------------------|----------------------|--------------------|-------------------|------------------------|--------------------|
> | WeiAvg and PFA                                 |  **4.24** (32.95)     | **4.11** (30.68)   |  4.95 (25.01)     | **3.89** (**27.84**)    |5.70 (17.04)         |
> | DPFedAvg  and minimum $\epsilon$ |  4.92 (**34.69**)     | 4.71 (29.54)        |  4.95 (25.41)     | 6.78 (14.77)              |6.86 (11.36)         |
> | Robust-HDP                                       |  4.77 (34.09)          | 4.59 (**34.91**)   |  **4.66** (**26.13**) | 6.01 (15.34)    | **3.89** (**18.97**)         |
>
> We observe that when we have homogeneous $b_i$ and $\epsilon_i$ and heterogeneous $N_i$, using Robust-HDP results in more fairness in the system, as it assigns more balanced weights to clients in this case.
>
>
> > **Comment2**: The experimental results shown in the body part are mainly on MNIST and FMNIST.
>
> **A2**: Following your comment and other reviewers, we have added multiple experimental results on CIFAR10 and also with an even larger model **ResNet-34** to evaluate Robust-HDP in more complex cases. Experimental results still show the superiority of Robust-HDP. Please read our answer **A2** to reviewer **fjuK** for results in more complex settings.
>
>
> Please continue to our second official comment in the following:

---

> ### Author Response · Authors · 2023-11-22
> **Response to Reviewer eRk3**
>
> > **Comment3**: Also, the results of CIFAR10 seem to be lost in the Appendix
>
> **A3**: By mistake, we did not include the table of detailed results for CIFAR10 in the appendix at the submission time. We have now included it in the following:
>
> | Algorithm     |  Dist1  |  Dist2 |  Dist3  | Dist4  | Dist5 | Dist6 | Dist7  | Dist8 | Dist9 |
> |---------------|---------|--------|---------|--------|--------|--------|--------|------|--------|
> | WeiAvg        |  31.18 |31.18  |29.65  |27.74  |24.25  |19.91 | 21.93  |18.91 | 20.64 |
> | PFA            |  26.91| **32.68** |25.19| 29.21 |21.63 |18.93 |20.63 |16.27 |15.75 |
> | DPFedAvg    |  31.51  |21.51 | 22.28  |20.50  |21.25  |15.19 | 18.27 | 16.45 | 18.63 |
> | minimum $\epsilon$ | 26.20 | 16.71 | 16.45 | 15.86 | 14.23 | 10.51 | 13.35 | 13.32 | 14.11|
> | Robust-HDP |   **31.97**  |31.70 | **32.0** | **30.60** | **24.86** | **23.61** | **24.1**  |**19.02** | **22.05** |
>
> > **Q1**: I do not find how the data heterogeneity is simulated in the experiments
>
> **A4**: In section A.1 of the appendix (DATASETS AND MODELS), we have explained in details how we distribute data across clients. We first split each existing class into some shards. Then, depending on the desired level of data heterogeneity, we decide about the maximum number of classes that each client can have samples from. This results in uniform dataset size across clients. In cases where we want to have quantity shift, we use Dirichlet allocation.
>
>
> > **Q2**: the server can eliminate the noise from the model and get the noise-free results of the local model update
>
> **A5**: Based on post-processing property, it's always safe to perform arbitrary computations on the output of a differentially private mechanism, so it is OK to use the matrix $L$ at aggregation time. However, our results show that using $L$ is not better than using  $M$  with the aggregation weights coming from Robust-HDP.

---

### Official Review · Reviewer_fjuK · 2023-10-30

**Soundness:** 3 good
**Presentation:** 2 fair
**Contribution:** 3 good
**Rating:** 6
**Confidence:** 3

**Summary:**

This paper is about using DP in conjunction with FL (DPFL). The key insight is that if we heterogeneity in DP (via different privacy requirements) and heterogeneity in FL (with different types of devices, with different amounts of memory), approaches that deal with only one of these types of heterogeneity are sub-optimal when both occur simultaneously.

This paper introduces a new algorithm that considers both of these concerns simultaneously, using a robust PCA like approach.

**Strengths:**

1. Identifies a problem that is important to solve.

2. Presents an interesting solution

3. The simulation results are convincing, albeit only for small models.

**Weaknesses:**

1. Presentation is poor, with quite a few formatting errors --- please address these.

2. I don't really see anything too meaningful in the theoretical results. The idea that the procedure converges is good, but the result is too much of a mess to really interpret. Can this be cleaned up?

3. There are no simulation results where the two types of heterogeneity are homogeneous. I would like to see these, and see if your algorithm is outperformed in these cases

4. Simulations are on MNIST, CFAR10 size datasets/models. Time and time again, we have seen that insights at this scale are not generalizable. I would like to see at least one example on a bigger model, even if it is just fine tuning.

**Questions:**

1. An interesting question is incentives. Does your approach lead users to lie about their memory constraints to get free extra privacy? It seems to me that this might be the case. Some suggested references for this include

[1] Fallah, Alireza, et al. "Optimal and differentially private data acquisition: Central and local mechanisms." Operations Research (2023).

[2] Donahue, Kate, and Jon Kleinberg. "Model-sharing games: Analyzing federated learning under voluntary participation." Proceedings of the AAAI Conference on Artificial Intelligence. Vol. 35. No. 6. 2021.

[3] Kang, Justin, Ramtin Pedarsani, and Kannan Ramchandran. "The Fair Value of Data Under Heterogeneous Privacy Constraints." arXiv preprint arXiv:2301.13336 (2023).

[4] Karimireddy, Sai Praneeth, Wenshuo Guo, and Michael I. Jordan. "Mechanisms that incentivize data sharing in federated learning." arXiv preprint arXiv:2207.04557 (2022).

I would also like to see some additional simulations:

2. Simulations where $\epsilon_i$ and $b_i$ are the same for all $i$, to see if existing approaches win, and explain why

3. Larger scale models in simulations

---

> ### Author Response · Authors · 2023-11-22
> **Response to Reviewer fjuK**
>
> Thank you for your feedbacks and comments about our work. We are happy to answer your comments and questions:
> > **Q1**: Does your approach lead users to lie about ...?
>
> **A1**: The answer is no. Because our algorithm does not operate based on clients privacy parameters $\{\epsilon_i\}$. Whether the server knows $\\{\epsilon_i\\}_i$ or not does not affect Robust-HDP, because it operates based on estimating the amount of noise in clients model updates $\\{\sigma_i^2\\}_i$. However WeiAvg/PFA do heavily depend on the server knowing clients privacy parameters. Due to this dependence , a client could also lie about their $\epsilon_i$ and mislead the server, which runs WeiAvg/PFA, by pretending that they are a less privacy sensitive client by sending a larger $\epsilon$ .
>
>
> > **Q2**: Simulations where $\epsilon_i$ and $b_i$ are uniform.
>
> **A2**:  We ran a set of experiments with different assumptions. Unlike before where we assumed heterogeneous $\epsilon_i$, heterogeneous $b_i$ and uniform dataset size $N_i$, we experimented with the following cases. **As asked in your comments, all the the experiments are run on CIFAR-10** (instead of MNIST/FMNIST), which uses ResNet18 with $11, 181, 642$ parameters:
>
> **case 1.** uniform $\{bi = 32\}$, heterogeneous  $\epsilon_i$ and heterogeneous $N_i$. Results for homogeneous $b_i=b=32$ and heterogeneous $\epsilon_i$ on CIFAR10, still shows superiority of Robust-HDP:
>
> | Algorithm     |  Dist1  |  Dist2 |  Dist3  | Dist4  | Dist5 | Dist6 | Dist7  | Dist8 | Dist9 |
> |---------------|---------|--------|---------|--------|--------|--------|--------|------|--------|
> | WeiAvg        |  31.99  | 31.18 |  29.59 | 25.97  | 24.66 | 16.61 | **23.13** | 17.01 | 14.34 |
> | PFA            |  30.89  | 32.02 |  28.34 | 24.17  | 22.23 | 15.31 | 22.15 | 16.32 | 13.11 |
> | DPFedAvg    |  33.89  | 24.16 |  24.62 | 17.50  | 22.71 | 16.76 | 19.52 | 13.81 | **16.27** |
> | Robust-HDP |  **34.94**  | **33.78** |  **31.34** | **32.50**  | **26.05** | **17.98** | **23.13** | **17.97** | 15.79 |
>
> **case 2.** heterogeneous $\{bi\}$, uniform $\epsilon_i=\epsilon$ ($\epsilon$ is fixed to mean of distributions 1, 3, 5 and 7), and heterogeneous $N_i$ (quantity shift):
>
> | Algorithm                                            |  $\epsilon=2.6$  |  $\epsilon=2$ |  $\epsilon=1.1$  | $\epsilon=0.6$  | $\epsilon=0.35$ |
> |-------------------------------------------|--------------------|-----------------|--------------------|--------------------|--------------------|
> | WeiAvg and PFA                                 |  35.86                 | 33.50             |  29.21                  | **24.49**                 | 18.40                  |
> | DPFedAvg  and minimum $\epsilon$  |  37.00                | 32.89             |  29.32                  | 23.06                     | 19.14                   |
> | Robust-HDP                                        |  **37.45**            | **34.93**        |  **29.78**             | 23.15                    | **19.54**                |
>
> **case 3.** uniform $\{bi\}$, uniform $\epsilon_i=\epsilon$ and heterogeneous $N_i$:
> this is the regular homogeneous DPFL, **for which DPFedAvg is designed**. Based on equations 7, 5 and 3, $\sigma_i^2$ changes across clients just based on their dataset size $N_i$. However, $N_i$ appears twice in eq.7 with different effects. Hence $\sigma_i^2$ varies very slowly with $N_i$ (we have experimentally confirmed this). Therefore, Robust-HDP (which solves problem 8) and WeiAvg ($w_i \propto \epsilon_i$) assign uniform aggregation weights $w_i$ to all clients. However, DPFedAvg assigns larger weights to clients with larger dataset size ($w_i \propto N_i$). **As all clients have the same  $\sigma_i^2$ in this case, it makes sense to assign larger weights to the clients with larger dataset sizes (as DPFedAvg does) to improve utility**. But what happens to fairness in the system? it has been shown that accuracy of DP models drops much more for the underrepresented subgroups [1].  Hence, in some scenarios, we expect that Robust-HDP improves fairness in the system for clients with lower $N_i$. For instance, the following results evaluate the utility and also fairness in the system when we have homogeneous DPFL and only $N_i$ varies across clients. We have looked at std of clients test accuracies and also the test accuracy of the client with smallest dataset size ($N_i$) (in parentheses).
>
> [1]: E. Bagdasaryan, "Differential Privacy Has Disparate Impact on Model Accuracy", arXiv 2019.
>
> Please continue to the our second official comment below:

---

> ### Author Response · Authors · 2023-11-22
> **Response to Reviewer fjuK**
>
> * utility (average test accuracy). Note that PFA and WeiAvg are equivalent when $\\{\epsilon_i\\}_i$ is uniform:
> | Algorithm                                            |  $\epsilon=2.6$  |  $\epsilon=2$ |  $\epsilon=1.1$  | $\epsilon=0.6$  | $\epsilon=0.35$ |
> |-------------------------------------------|--------------------|-----------------|--------------------|--------------------|--------------------|
> | WeiAvg and PFA                                 |  37.24                 | **34.90**        |  27.80                 | 23.22                  | 19.01                  |
> | DPFedAvg  and minimum $\epsilon$ |  **38.52**            | 32.78             |  **30.42**            | **23.50**             | 20.26                 |
> | Robust-HDP                                        |  37.68.               | 32.54.            |  27.51                 | 22.58                   | **20.52**            |
>
>
> * performance fairness, measured by std of clients test accuracies and also the test accuracy of the client with smallest dataset size ($N_i$) (in parentheses):
> | Algorithm                                            |  $\epsilon=2.6$    |  $\epsilon=2$     |  $\epsilon=1.1$ | $\epsilon=0.6$        | $\epsilon=0.35$ |
> |-------------------------------------------|----------------------|--------------------|-------------------|------------------------|--------------------|
> | WeiAvg and PFA                                 |  **4.24** (32.95)     | **4.11** (30.68)   |  4.95 (25.01)     | **3.89** (**27.84**)    |5.70 (17.04)         |
> | DPFedAvg  and minimum $\epsilon$ |  4.92 (**34.69**)     | 4.71 (29.54)        |  4.95 (25.41)     | 6.78 (14.77)              |6.86 (11.36)         |
> | Robust-HDP                                       |  4.77 (34.09)          | 4.59 (**34.91**)   |  **4.66** (**26.13**) | 6.01 (15.34)    | **3.89** (**18.97**)         |
>
> We observe that when we have homogeneous $b_i$ and $\epsilon_i$ and heterogeneous $N_i$, using Robust-HDP results in more fairness in the system, as it assigns more balanced weights to clients in this case.
>
>
> * **Conclusion:** Based on the three experimental cases above, when there is heterogeneity in either $\\{b_i\\}_i$ or $\\{\epsilon_i\\}_i$ (or both), using Robust-HDP is beneficial, because in these cases $\\{\sigma_i^2\\}_i$ is also heterogeneous (according to eqs 7, 5 and 8). However, when both $b_i$ and $\epsilon_i$ are uniform (i.e. regular homogeneous DPFL), then $\sigma_i^2$ is almost fixed across clients and it makes sense to use DPFedAvg to assign larger weights to clients with larger dataset sizes $N_i$ and improve utility.
>
>
> > **Q3**: Larger Scale models in simulations.
>
> **A3**: We ran experiments on CIFAR10 with ResNet34 with $p=21,272,778$. Due to lack of time, we could run these experiments only with heterogeneous $b_i$ and heterogeneous $\epsilon_i$ (sampled from Dist3) and $E=100$ rounds:
>
> | Algorithm     |  Dist3  |
> |---------------|---------|
> | WeiAvg        |  18.26  |
> | PFA              |  15.49  |
> | DPFedAvg    |  15.47  |
> | Robust-HDP |  **22.67**  |

---

### Official Review · Reviewer_bWRm · 2023-10-30

**Soundness:** 2 fair
**Presentation:** 2 fair
**Contribution:** 2 fair
**Rating:** 5
**Confidence:** 3

**Summary:**

This paper studies federated learning with differential privacy guarantees. In particular, it focuses on the challenging problem where each client has different privacy budget and they may not want to share this information. The Robust-HDP algorithm is proposed, which features robust PCA in the aggregation step. Both theoretical and empirical evidence are provided.

**Strengths:**

The problem under consideration is of great importance and well motivated in the paper. The writing is clear and empirical results are promising.

**Weaknesses:**

In my view, the main novelty of this paper is the use of RPCA in the aggregation step. Its main purpose is to allow the (untrusted) server to estimate the optimal weights without the knowledge of $(\epsilon_i,\delta_i)$. However, I have a few concerns:

1. The result stated in Lemma 1 seems to be independent of the use of RPCA. The bound in Lemma 1 only depends on $r$ and $\alpha_j$. Does the use of RPCA lead to explicit forms of these parameters?
2. I can understand that the matrix $M$ can be deposed into a low rank signal component $L$ and a noise component $S$, but why should one believes that $S$ is sparse?
3. The authors also mention that $\sigma_j^2 \gg 1$. Could the author further clarify this point? I don't quite understand the sentence in the parentheses "it is the noise power..." and the terms in (3) and (5) both look like $o(1)$ to me, at least when $N_i$ is large.

**Questions:**

In addtion to the above, I believe a definition of Var is needed as I don't see how it maps from a vector to a scalar (e.g.\ in (3) and (5)).

---

> ### Author Response · Authors · 2023-11-22
> **Response to Reviewer bWRm**
>
> Thank you for your feedbacks and comments about our work. We are happy to answer your comments and questions:
> > **Comment1**: The result stated in Lemma 1 seems to be independent of the use of RPCA.
>
> **A1**: Lemma1 says that if RPCA can extract the matrix $S$ up to a factor $r$, then Robust-HDP can estimate the  clients noise variances $\\{\sigma_i^2\\}_i$ up to a factor $r^2$. In ideal case $r=1$ and $\alpha_j=0$. Based on the theoretical results in the RPCA paper, as rank of $L$ decreases and the additive noisy matrix $S$ gets more sparse (which we have explained in the following why this is the case), we get closer to the ideal case of extracting $S$ exactly. Therefore, the parameters $r$ and $\alpha_j$ depend on the given matrix $M$ and the RPCA algorithm. On the other hand, for our purpose of estimating the optimum weights $w_i^*$, estimating the elements in $S$ up to a factor $r$ is sufficient (so for our goal, $r$ does not need to be necessarily $1$). As an instance for MNIST and Dist8, we can see from Figure 4 that $\frac{\hat{\sigma}_i^2}{\sigma_i^2} \approx r^2$ is very close to 1 for all the 20 clients in the system. This means that for MNIST that $M$ has columns of dimension $p=28,939$ (number of model parameters), RPCA can estimate $S$ with a good precision, which consequently results in the estimates $\\{\hat{\sigma}_i^2\\}_i$ being very close to their true values $\\{\sigma_i^2\\}_i$.
>
> Now consider the case where $p$ is too large, e.g. for the ResNet18 ($p=11,181,642$) or ResNet34 ($p=21,272,778$).  Then running RPCA on the matrix $M$, not only takes long time, but also it might result in lower precision in extracting $S$. Hence, we proposed our approximation approach relying on two facts: **1. in our heterogeneous DPFL problem, all the components in $S_{:,i}$ are iid, because they were sampled and added by the same client $i$ during its local DP training  2. We want to estimate $\sigma_i^2 = ||S_{:,i}||^2= \sum_{l=1}^p Sl,i^2$, which is the sum of the variances of the iid samples $\\{Sl,i\\}_{l=1}^p$.** As sample variance decreases inversely proportional to the number of samples, even if we estimate $\sigma_i^2$ by running RPCA on a submatrix of $M$ with $1\ll p' \ll p$ (e.g. $p'=200,000$ that we used), we still get to an approximation of $\sigma_i^2$ up to a factor $\frac{p'}{p}$, which is sufficient for Robust-HDP to get to the optimum weights $\\{w_i^*\\}_i$. Our multiple improved experimental results on CIFAR10 shows that the approximation idea is indeed effective.
>
>
>
> > **Comment2**: I can understand that the matrix $M$ can be deposed into a low rank signal component $L$  and a noise component $S$, but why should one believes that $S$ is sparse?
>
> **A2**: As shown in Figure 1, **$\sigma_i^2$, which is equal to $||S_{:,i}||^2$**, heavily depends on $\epsilon_i$ and $b_i$. When both $\epsilon_i$ and $b_i$ are small, $\sigma_i^2$ increases fast, and in other cases it has relatively smaller values. The heterogeneity in clients privacy parameters $\epsilon_i$ and batch sizes $b_i$ results in a variation between $\{\sigma_i^2\}$ across clients, which **makes the sparse patterns similar to what observed in Figure 4 or Figure 10  (in the appendix) for $\sigma_i^2$**. Now note that  a few of existing clients might be in the peak corner of Figure 1. Hence, the pattern in $\\{\sigma_i^2=||S_{:,i}||^2\\}$ is sparse. This means that some columns of $S$ have large norm and some others have small norms. Now, note that the elements in $\mathbf{S}_{:,i}$ are iid with mean 0 (they are noise elements), this means that $\mathbf{S}$ is also sparse, or more precisely most columns of $S$ hold elements with small magnitude and some other columns have elements with relatively larger magnitudes. Hence $S$ is can be thought of as being sparse (again see the pattern in Fig. 4 of the norms of the 20 columns of $S$ ).
>
> > **Comment3**:The authors also mention that $\sigma_j^2 \gg 1$.
>
> **A3**:  Based on the definition above, $\sigma_j^2$ is the noise variance in the whole model update of dimension $p$. It can be computed from eq.7, which is computed based on either eq.3 or eq.5. In either case, the terms in equations 3 and 5 are large, **mainly because $p$ which is the number of model parameters, has a large value** (e.g. 11,181,642 for CIFAR10 or 28,938 for MNIST/FMNIST). Also $c$ (clipping threshold), and $z$ (the DP noise scale) are constants around 1. In order to get an idea of the values of $\sigma_i^2$, see the values in Figure 4 left, which shows the true noise variance existing in each of the 20 clients model updates.
>
>
> > **Q1**: I believe a definition of Var is needed as I don't see how it maps from a vector to a scalar (e.g.\ in (3) and (5)).
>
> **A4**: First, for a vector $V$ of dimension $p$ we have: **$\texttt{Var}(V) = \sum_{I=1}^p \mathbb{E}[(v_i - \mathbb{E}[v_i])^2]$**. So it captures the deviations of $V$ from its mean. Note than when $v_i$ are iid and $p$ increases, $\texttt{Var}(V)$ also increases.

---

### Official Review · Reviewer_ThZo · 2023-10-31

**Soundness:** 3 good
**Presentation:** 2 fair
**Contribution:** 2 fair
**Rating:** 5
**Confidence:** 4

**Summary:**

The paper proposes a method for weighting client updates in differentially private (DP) federated learning (FL), where the clients can have differing local DP guarantees. The main idea is to have the server estimate the total noise level of each client update via robust PCA, and weight the clients in federated averaging accordingly, giving more weight to clients with less noisy updates. The authors then show experimentally that their weighting method improves model utility compared to existing alternatives in the presence of heterogeneous noise levels on the clients (differing DP guarantees, different subsampling fractions).

**Strengths:**

* While there are existing works proposing to weight client updates according to heterogeneous privacy preferences, using estimated noise values directly instead of e.g. simply epsilon values seems like a good idea.

* The authors show convergence results under some assumptions.

* The paper is mostly fairly easy to read.

* An effective way for optimising weights in fedavg to improve utility is an important research direction.

**Weaknesses:**

* As the authors note, as the proposed method involves the server running robust PCA on the model updates, it can only scale via approximating the proposed method. It is not clear from the theory or from the provided empirical results how feasible the approximation approach actually is (in terms of approximation quality vs compute).

* E.g. end of Sec.3.2: having the clients keep their noise addison mechanisms as well as epsilon budget secret (which is one of the main assumptions used to motivate the mechanism-oblivious noise estimation method) is very much not standard in DP. I do not find the argument all that convincing: since the standard DP guarantees assume that the privacy budget is open information, ideally I would very much think that a rational client would simply add enough noise to be comfortable with the amount of information it is sharing, instead of using less noise than they are actually okay with and trying to keep the privacy budget secret to get some unquantifiable amount of extra privacy.

* Parts of the presented theory, e.g., long analysis of the connection between noise level and batch size seem quite disconnected from the actually proposed method: as far as I see, using robust PCA and then weight the updates according to the estimated noise level does not actually rely on the batch size analysis in any significant way beyond getting some motivation for having differing noise levels on the clients.

**Questions:**

1) Sec.1, : on clients having to use small batch size due to memory constraints: I would think that using simple grad accumulation alleviates the memory problem, and from Fig1 one could think that quite modest increase in batch sizes suffice to reduce the noise level significantly. Is there some specific reason why this would not work for the case you consider? Somehow framing the discussion around client memory limitations seems a bit weird.

2) Sec.3.3: "all the clients are training the same shared model on more or less similar data distributions". How sensitive the rank of M is for client heterogeneity?


3) In eq.3: any reasoning about when the final approximation is good?

4) Please mention the neighbourhood relation explicitly in defining DP.

#### Minor comments/typos etc (no need to comment)

* Sec.4.2, RQ1; Fig.2 caption have some broken references.

* Eq.(3): is Var element-wise variance? Please mention to avoid some confusion, also the notation changes somewhat confusingly between text and appendix, e.q. p vs d for dimensionality

---

> ### Author Response · Authors · 2023-11-22
> **Response to Reviewer ThZo**
>
> Thank you for your feedbacks and comments.
>
>
> > **Q1**: framing the discussion around client memory limitations seems a bit weird.
>
> **A1**: Even with gradient accumulation, the amount of DP noise in clients updates, depends on the base batch size that they use. The accumulation of $m$ batch gradients with base batch size $b_i$, which is used for model updates is: $\\sum_{j=1}^m \bigg( \frac{1}{b} \bigg [ \big (\sum_{i \in \mathcal{B}_j} \bar{g}_i(\theta) \big) + \mathcal{N}(0, \sigma^2 I_p) \bigg]\bigg),$ where $\sigma = c.z(\epsilon_i, \delta_i, \mathbf{\frac{b_i}{N_i}}, K_i, E)$ depends on the **base sampling ratio $\frac{b_i}{N_i}$** that the client uses for computing batch gradients.  Therefore, although the stochastic noise in accumulated batch gradient decreases, the total DP noise induced in the model update of each client after $K_i$ local epochs does not change (compared to when we use batch size $b_i$). Hence heterogeneity in the base batch sizes $b_i$ results in heterogeneity in $\sigma_i^2$. This being said, consider when all clients use the same batch size $b_i=b$, but their privacy parameters are heterogeneous. **Based our analysis in eq. 7, 5 and 3**, this results in the induced noise variances $\sigma_i^2$ being heterogeneous. This makes WeiAvg a suboptimal solution. In contrast Robust-HDP assigns aggregation weights **based on the real noise variance in clients model updates ($\{\sigma_i^2\}$)**, instead of their privacy parameters $\{\epsilon_i\}$ (which might not even be shared with the server). Also, **for Robust-HDP does not matter if $\epsilon_i$ are known by server or not, because it does not use them, so it gives clients more "freedom" for not sharing them with the server**. Results for homogeneous $b_i=b=32$ and heterogeneous $\epsilon_i$ on CIFAR10, still shows superiority of Robust-HDP:
>
> | Algorithm     |  Dist1  |  Dist2 |  Dist3  | Dist4  | Dist5 | Dist6 | Dist7  | Dist8 | Dist9 |
> |---------------|---------|--------|---------|--------|--------|--------|--------|------|--------|
> | WeiAvg        |  31.99  | 31.18 |  29.59 | 25.97  | 24.66 | 16.61 | **23.13** | 17.01 | 14.34 |
> | PFA            |  30.89  | 32.02 |  28.34 | 24.17  | 22.23 | 15.31 | 22.15 | 16.32 | 13.11 |
> | DPFedAvg    |  33.89  | 24.16 |  24.62 | 17.50  | 22.71 | 16.76 | 19.52 | 13.81 | **16.27** |
> | Robust-HDP |  **34.94**  | **33.78** |  **31.34** | **32.50**  | **26.05** | **17.98** | **23.13** | **17.97** | 15.79 |
>
> > **Q2**: How sensitive the rank of M is for client heterogeneity?
>
> **A2**: We have run some experiments on MNIST and instead of assigning all classes to all (20) clients, we have limited the number of classes per client to be at maximum 8, which results in more data heterogeneity. Note that RPCA needs the matrix $M$ to have "low rank" (not rank 1). So as long as the data heterogeneity across clients results in a low rank $M$, RPCA and consequently Robust-HDP can be used to improve utility, especially when $\epsilon_i$ are small:
>
> | Algorithm     |  Dist1  |  Dist2 |  Dist3  | Dist4  | Dist5 | Dist6 | Dist7  | Dist8 | Dist9 |
> |---------------|---------|--------|---------|--------|--------|--------|--------|------|--------|
> | WeiAvg        |  **90.24**  | 83.43 |  **89.34** | 88.28  | **87.43** | 77.86 | 81.45 | 76.73 |  79.72|
> | PFA            |  82.98  | 88.69 |  82.14 | 85.67  | 81.07 | 77.44 | 71.90 | 65.43 | 69.14 |
> | DPFedAvg    |  88.45  | 81.76 |  80.72 | 80.04 |79.17  | 75.30 | 82.60 | 70.82 | 78.39 |
> | Robust-HDP |  87.66  | **88.60** | 86.8  | **88.60**  | 83.20 |**80.19** | **83.05** | **78.40** | **81.14** |
> .
>
> >**Q3**: In eq.3: any reasoning about when the final approximation is good?
>
> **A3**: First, for a vector $V$ of dimension $p$ we have: **$\texttt{Var}(V) = \sum_{I=1}^p \mathbb{E}[(v_i - \mathbb{E}[v_i])^2]$**. From the values that we had in our experiments, the first term in eq.3 is less than 1 (similarly $\frac{c^2}{b_i^2}$ in the second term is smaller than 1 too): because $c$ is set to 3 in our experiments and $b_i$ is a random value from $\\{16, 32, 64, 128\\}$. On the other hand $p$ (the number of model parameters) is large (e.g. 11,181,642 for CIFAR10 or 28,938 for MNIST/FMNIST) which makes the second term much larger than the first term (look at numbers in Figure 4). Hence, the approximation makes complete sense.
>
> >**Q4**: Please mention the neighbourhood relation explicitly in defining DP.
>
> **A4**: Two datasets $\mathcal{D}_i$ and $\mathcal{D}'_i$ (for client $i$) are adjacent datasets, if they differ in only one data sample (present in one dataset and absent in the other one). Therefore, our notion of adjacency is sample level.
>
> >**Q5**: Fig.2 caption have some broken references.
>
> Please continue to the our second official comment below:

---

> ### Author Response · Authors · 2023-11-22
> **Response to Reviewer ThZo**
>
> >**Q5**: Fig.2 caption have some broken references.
>
> **A5**: By mistake, we did not include the table of detailed results for CIFAR10 in the appendix at the submission time. We have now included it in the following:
>
> | Algorithm     |  Dist1  |  Dist2 |  Dist3  | Dist4  | Dist5 | Dist6 | Dist7  | Dist8 | Dist9 |
> |---------------|---------|--------|---------|--------|--------|--------|--------|------|--------|
> | WeiAvg        |  31.18 |31.18  |29.65  |27.74  |24.25  |19.91 | 21.93  |18.91 | 20.64 |
> | PFA            |  26.91| **32.68** |25.19| 29.21 |21.63 |18.93 |20.63 |16.27 |15.75 |
> | DPFedAvg    |  31.51  |21.51 | 22.28  |20.50  |21.25  |15.19 | 18.27 | 16.45 | 18.63 |
> | minimum $\epsilon$ | 26.20 | 16.71 | 16.45 | 15.86 | 14.23 | 10.51 | 13.35 | 13.32 | 14.11|
> | Robust-HDP |   **31.97**  |31.70 | **32.0** | **30.60** | **24.86** | **23.61** | **24.1**  |**19.02** | **22.05** |
>
>
> > **comment1**: It is not clear from the theory or from the provided empirical results how feasible the approximation approach actually is.
>
> **A6**: Lemma1 says that if RPCA can extract the matrix $S$ up to a factor $r$, then Robust-HDP can estimate the  clients noise variances $\\{\sigma_i^2\\}_i$ up to a factor $r^2$. In ideal case $r=1$ and $\alpha_j=0$. Based on the theoretical results in the RPCA paper, as rank of $L$ decreases and the additive noisy matrix $S$ gets more sparse, we get closer to the ideal case of extracting $S$ exactly. Therefore, the parameters $r$ and $\alpha_j$ depend on the given matrix $M$ and the RPCA algorithm. On the other hand, for our purpose of estimating the optimum weights $w_i^*$, estimating the elements in $S$ up to a factor $r$ is sufficient (**so for our goal, $r$ does not need to be necessarily $1$**). As an instance for MNIST and Dist8, we can see from Figure 4 that $\frac{\hat{\sigma}_i^2}{\sigma_i^2} \approx r^2$ is very close to 1 for all the 20 clients in the system. This means that for MNIST that $M$ has columns of dimension $p=28,939$ (number of model parameters), RPCA can estimate $S$ with a good precision, which consequently results in the estimates $\\{\hat{\sigma}_i^2\\}_i$ being very close to their true values $\\{\sigma_i^2\\}_i$.
>
> Now consider the case where $p$ is too large, e.g. for the ResNet18 ($p=11,181,642$) or ResNet34 ($p=21,272,778$).  Then running RPCA on the matrix $M$, not only takes long time, but also it might result in lower precision in extracting $S$. Hence, we proposed our approximation approach relying on two facts: **1. in our heterogeneous DPFL problem, all the components in $S_{:,i}$ are iid, because they were sampled from the same normal distribution and added by the same client $i$ during its local DP training  2. We want to estimate $\sigma_i^2 = ||S_{:,i}||^2= \sum_{l=1}^p Sl,i^2$, which is the sum of the variances of the iid samples $\\{Sl,i\\}_{l=1}^p$.** As sample variance decreases inversely proportional to the number of samples, even if we estimate $\sigma_i^2$ by running RPCA on a submatrix of $M$ with $1\ll p' \ll p$ (e.g. $p'=200,000$ that we used), we still get to an approximation of $\sigma_i^2$ up to a factor $\frac{p'}{p}$, which is sufficient for Robust-HDP to get to the optimum weights $\\{w_i^*\\}_i$. Our multiple improved experimental results on CIFAR10 shows that the approximation idea is indeed effective.

---

### Author Response · Authors · 2023-11-22
**Thank you all for your comments/questions**

Dear reviewers,

As the author-reviewer discussion deadline is approaching, we would like to take the chance to thank you all for your comments and questions. The study and experiments that we conducted for answering them enhanced our understanding of our proposed algorithm and why it works too. **We encourage each of you to read questions of other reviewers and our answers to them too, as your questions have been quite related**. We sincerely appreciate the time and effort you put in this draft.